# FFN Fusion: Rethinking Sequential Computation in Large Language Models

**Akhiad Bercovich**[*]     **Mohammad Dabbah**[*]     **Omri Puny**[*]

**Ido Galil**     **Amnon Geifman**     **Yonatan Geifman**     **Izhak Golan**     **Ehud Karpas**

**Itay Levy**     **Zach Moshe**     **Najeeb Nabwani**     **Tomer Ronen**     **Itamar Schen**

**Elad Segal**     **Ido Shahaf**     **Oren Tropp**     **Ran Zilberstein**     **Ran El-Yaniv**

NVIDIA, {abercovich, mdabbah, opuny, relyaniv}@nvidia.com

## Abstract

We introduce *FFN Fusion*, an architectural optimization technique that reduces sequential computation in large language models by identifying and exploiting natural opportunities for parallelization. Our key insight is that sequences of Feed-Forward Network (FFN) layers, particularly those remaining after the removal of specific attention layers, can often be parallelized with minimal accuracy impact. We develop a principled methodology for identifying and fusing such sequences, transforming them into parallel operations that significantly reduce inference latency while preserving model behavior. Applying these techniques to Llama-3.1-405B-Instruct, we create Llama-Nemotron-Ultra-253B-Base (Ultra-253B-Base), an efficient model that achieves a $1.71\times$ speedup in inference latency and $35\times$ lower per token cost while maintaining strong performance across benchmarks. Most intriguingly, we find that even full transformer blocks containing both attention and FFN layers can sometimes be parallelized, suggesting new directions for neural architecture design.

## 1 Introduction

Large language models (LLMs) have emerged as one of the most transformative technologies of our time, revolutionizing how we approach artificial intelligence and computation. From powering sophisticated virtual assistants [16] to enabling breakthrough scientific research [1, 41], these models have evolved from academic curiosities [35] into indispensable tools that are reshaping entire industries. This transformation has been driven by rapid scaling to hundreds of billions of parameters [33, 2, 9], enabling unprecedented capabilities in reasoning, generation, and complex problem-solving [5, 15]. However, this extraordinary growth in scale and capability comes at a critical cost: the computational demands of running these models have become a fundamental bottleneck. As these models push the boundaries of what is possible with artificial intelligence, their deployment costs and resource requirements severely limit their accessibility, creating an urgent need for innovations that can make their capabilities widely available.

Research in LLM runtime optimization explored diverse approaches to managing computational demands. Traditional techniques like quantization [7, 12, 8, 47]—which reduce memory footprint and accelerate inference through lower-precision arithmetic—and pruning [26, 18, 17, 28, 11]—which

39th Conference on Neural Information Processing Systems (NeurIPS 2025).

removes redundant parameters—have become standard tools. A more recent innovation, Mixture-of-Experts (MoE) [38, 23], has demonstrated remarkable potential by dynamically activating only a small subset of model parameters during inference. The DeepSeek-V3 architecture [6] pushes this approach to new extremes, employing 256 expert FFN modules at each layer while activating only 8 experts per token (plus one shared expert), effectively achieving the capabilities of a much larger model while maintaining reasonable computational costs through sparse activation. However, each of these approaches faces distinct challenges: quantization encounters precision-accuracy trade-offs at small bit numbers, pruning struggles to identify significant additional redundancy without compromising accuracy performance (and pruning can be harnessed to efficiency mainly when it is structured), and MoE architectures, while efficient for single-token inference or for very large batches, do not provide optimal throughput at small and intermediate batch sizes due to under-utilization of compute resources, as discussed in Appendix E. Given the above limitations, there is an urgent need for complementary approaches that can unlock new dimensions of efficiency improvement while maintaining the simplicity and predictable scaling characteristics of dense architectures.

In this work, we introduce two major contributions that fundamentally advance the state of LLM efficiency. First, we present FFN Fusion, a novel architectural optimization technique that challenges the conventional sequential nature of transformer computation, and is illustrated in Figure 1. By identifying and exploiting patterns of computational independence in FFN layers, our approach enables parallel execution across multiple GPUs while preserving model functionality. This parallelization is particularly effective on modern GPU nodes, where tensor-parallel implementations often suffer from synchronization delays between consecutive layers. By concentrating the computation into fewer layers and reducing cross-device communication, our method significantly improves hardware utilization. Our findings reveal that substantial portions of LLM computation can be parallelized with minimal accuracy impact, complementing existing runtime optimization techniques like pruning and quantization. Second, we demonstrate the practical impact of these insights through Ultra-253B-Base, a state-of-the-art 253B parameter model derived from Llama-3.1-405B-Instruct (henceforth Llama-405B), derived using FFN Fusion and attention pruning. This model not only showcases the scalability of our approach, but also achieves remarkable efficiency gains while maintaining or exceeding its parent model's capabilities across a comprehensive suite of benchmarks.

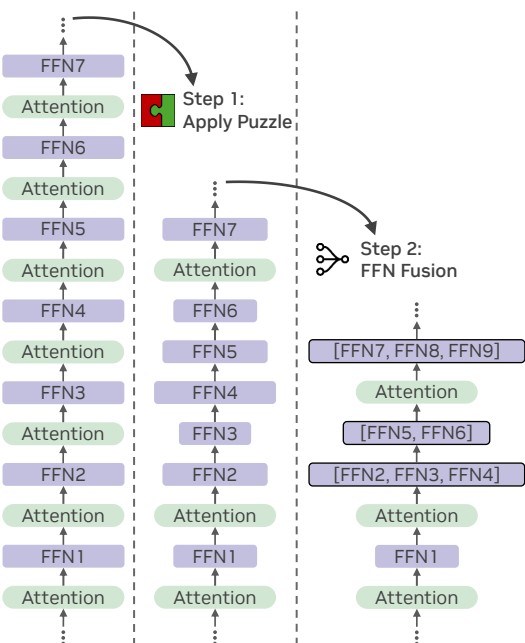

Figure 1: An overview of our FFN Fusion approach. Step 1: We apply Puzzle to partially remove FFN layers and remove entire attention layers. Step 2: We fuse consecutive FFN layers into a single wide FFN layer. Fusion is denoted using the bracket notation $\left[\text{FFN}^i, \ldots, \text{FFN}^{i+k}\right]$.

The applicability of FFN Fusion rests on a fundamental insight about modern LLM architectures. Recent work has shown that LLM display structural redundancy [43, 31, 10, 49, 36, 25, 14], particularly attention mechanisms, which can be selectively removed with minimal accuracy impact [3, 19], often leaving models with extended sequences of consecutive FFN layers. We demonstrate that these FFN sequences exhibit surprisingly low inter-layer dependencies, enabling a novel transformation: multiple sequential FFN layers can be fused into a single, wider layer that enables simple parallel execution. This transformation is particularly powerful, eliminating synchronization points while allowing for more efficient hardware utilization without requiring architectural redesign (as visualized in Figure 2). Through analysis and empirical validation, we argue for the conditions under which this fusion preserves model behavior, showing that these conditions are commonly satisfied in practice, especially in larger models where the potential efficiency gains are most significant. Most surprisingly, our preliminary investigations suggest that even complete transformer blocks—containing both attention and FFN components—can sometimes be parallelized, pointing to potential new directions in neural architecture design (see Appendix B). Leveraging these insights, we develop Ultra-253B-Base, derived from Llama-405B through a combination of attention pruning and FFN Fusion. This model achieves remarkable efficiency gains while maintaining or exceeding its parent model's capabilities:

- A $1.71\times$ speedup in inference latency and $35\times$ lower per-token cost at batch size 32.
- State-of-the-art performance on key benchmarks, including $84.92\%$ on Arena Hard, $86.58\%$ on HumanEval, $87.54\%$ on MMLU Instruct, $72.25\%$ on MMLU-Pro, and $9.19$ on MT-Bench.
- Improving memory footprint with a $2\times$ reduction in kv-cache memory, and reduced parameter count from 405B to 253B.

Our key contributions are:

- The discovery that some Feed-Forward Networks in transformer architectures can often be parallelized with minimal accuracy loss, challenging the conventional wisdom that sequential processing is necessary for these layers.
- The creation of Ultra-253B-Base, a powerful 253B parameter model that matches or exceeds Llama-405B's capabilities while offering substantially improved efficiency, to be publicly released upon acceptance.
- Preliminary evidence that even complete transformer blocks—containing both attention and FFN components—can sometimes be parallelized, opening new possibilities for architectural innovations in large language models.

## 2 Preliminaries

**Transformer-based LLMs.** The structure of LLMs is typically based on the widely used transformer architecture [42]. It is generally composed of a series of sequential blocks, denoted as $f^1, \ldots, f^m$, each containing an attention layer and an FFN layer. The attention layers capture the contextual relationships between input tokens, while the FFN layers apply transformations independently to each token. Given an input $\boldsymbol{X} \in \mathbb{R}^{n \times d}$, where $n$ represent the sequence length and $d$ the embedding dimension, a *transformer block* $f : \mathbb{R}^{n \times d} \to \mathbb{R}^{n \times d}$ is defined by the following equations:

$$g(\boldsymbol{X}) = \boldsymbol{X} + \text{Attention}(\eta_1(\boldsymbol{X})) \tag{1}$$

$$f(\boldsymbol{X}) = g(\boldsymbol{X}) + \text{FFN}(\eta_2(g(\boldsymbol{X}))), \tag{2}$$

where $\eta_k$ is a token-level normalization module, defined by the equation $\eta_k(x_i) = \frac{x_i}{\|x_i\|} \odot s_k$ with $x_i \in \mathbb{R}^d$ being the i-th token, $s_k \in \mathbb{R}^d$ being a learnable scale factor, and $\odot$ is the element-wise product. The FFN layer used in the models we experiment with is the *SwiGLU* module [37], which is defined as follows:

$$\text{SwiGLU}(\boldsymbol{X}) = (\sigma(\boldsymbol{X}W_2^T) \odot \boldsymbol{X}W_1^T)W_3^T, \tag{3}$$

where $\sigma$ is the SiLU activation [20] function, $W_1, W_2 \in \mathbb{R}^{d_h \times d_e}$ and $W_3 \in \mathbb{R}^{d_e \times d_h}$. Here, $d_e$ represents the embedding dimension, and $d_h$ denotes the hidden dimension of the FFN. Other components of the LLM include an embedding function at the model's input, which maps tokens to vectorized embeddings, an additional normalization function, and a linear function at the output, which normalizes the final block outputs and projects them to the vocabulary dimension.

**Puzzle.** Puzzle [3] is a neural architecture search (NAS) framework that optimizes a trained LLM for inference efficiency. Starting with a pre-trained model, it prunes or reconfigures each transformer block—often reducing or removing attention layers—while preserving model quality through a distillation process. In particular, applications of Puzzle often show that many attention layers can be removed with minimal accuracy loss, thus leaving sequences of FFNs uninterrupted by attention. Additionally, certain FFN layers experience significant channel pruning, leading to a reduction in their hidden dimension. Step 1 in Figure 1 illustrates a hypothetical outcome of this process.

## 3 FFN Fusion

Several optimization techniques reduce the number of attention layers to improve inference efficiency [3, 19]. An attention-removed transformer block is defined by the equation

$$\hat{f}(\boldsymbol{X}) = \boldsymbol{X} + \text{FFN}(\eta_2(\boldsymbol{X})). \tag{4}$$

Given a consecutive sequence of attention-removed blocks $\hat{f}^i, \ldots, \hat{f}^{i+c}$ we define the parallel version of this sequence $\hat{f}^{[i,i+c]}$ as

$$\hat{f}^{[i,i+c]}(\boldsymbol{X}) = \boldsymbol{X} + \sum_{j=0}^{c} \text{FFN}^{i+j}(\eta_2(\boldsymbol{X})), \tag{5}$$

where $\text{FFN}^i$ is the FFN layer of block $i$ and $\eta_2$ is taken from the last layer in the sequence. While in the sequential form the input for every FFN depends on the output of the previous layers, in this form all the FFN layers share the same input, which makes their computation independent and enables execution in parallel across different GPUs. Another useful property of this formulation is that equation 5 is equivalent to equation 4 with a single wider FFN when the weights are concatenated.

**Theorem 3.1.** *Let $n \in \mathbb{N}$, and let $FFN^1, \ldots, FFN^n$ be a sequence of FFN functions, where the weights of $FFN^i$ are $W_1^i, W_2^i, W_3^i$. Then, the sum of these FFN functions (equation 5) is equivalent to a single FFN function $FFN^*$, with the weight matrices given by:*

$$W_1^* = \left[ (W_1^1)^T, \ldots, (W_1^n)^T \right]^T$$
$$W_2^* = \left[ (W_2^1)^T, \ldots, (W_2^n)^T \right]^T$$
$$W_3^* = \left[ (W_3^1), \ldots, (W_3^n) \right],$$

*where $[\cdot, \ldots, \cdot]$ denotes the concatenation of matrices along the second axis and the dimensions of the matrices are $W_1^*, W_2^* \in \mathbb{R}^{nd_h \times d_e}$, $W_3^* \in \mathbb{R}^{d_e \times nd_h}$.*

The theorem (and proof in Appendix A) are written for the simple case where $d_h$ is equal for all the FFNs. The extension for the case where the hidden dimension is different for each layer is straightforward. While written in the context of Equation 3, the theorem holds for other common FFN variants [37, 39]. See a visualization in Figure 7.

**Efficiency motivation and analysis.** LLMs are ubiquitously designed in sequential blocks, with the block sizes and the number of blocks increasing as the models grow in size. For bigger models, parallelization techniques such as tensor parallel (TP) are utilized to split work across GPUs and reach acceptable inference latencies. However, this strategy does not result in linear latency improvements with the number of GPUs applied. The first reason, is the communication time required for the all–reduce that follows each TP block. The second reason is more subtle: GPUs are at their best for very large operations. As each atomic *General Matrix Multiplication* (GEMM) operation becomes smaller, low level overheads (GPU wave quantization) become more apparent and take a larger portion of the latency budget. Therefore, increasing the computation per GPU (by increasing the size of a block) while reducing the number of synchronizations needed (by using fewer blocks) is an effective strategy for low latencies in a TP setting (see Figure 2). Drawing from this motivation, we attempt to fuse many sequential blocks/FFNs in LLMs into fewer, larger blocks/FFNs. Each single reduction in the depth of the computational graph removes one unit of time spent on synchronization, and the bigger fused blocks also enable operating at higher TP numbers with enough computation assigned to each GPU. Figure 6 shows the effects described in this analysis using measurements of different model architectures in the same TP setting.

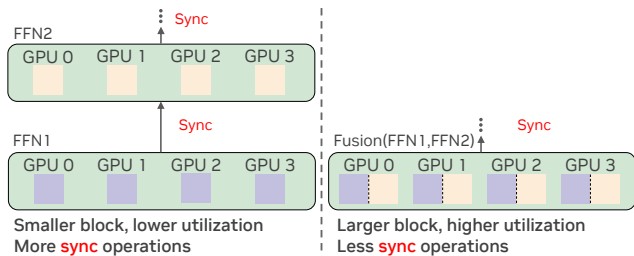

Figure 2: FFN Fusion helps reduce latency by increasing GPU utilization and by reducing syncs.

**Pairwise block dependency.** It is reasonable to hypothesize that not every transformer block in a sequential model is equally dependent on all its predecessors. In particular, a given block may depend only on a subset of the blocks that precede it. To investigate this, we perform a pairwise dependency analysis between blocks. Let us define $h(X) = f(X) - X$ as the contribution of $f$ to $X$, and similarly, $h^j$ as the contribution of $f^j$. We define $\tilde{h}_i^j$ as the contribution of block $j$ when block $i$ is removed from the model. We construct a dependency matrix $M \in \mathbb{R}^{m \times m}$ by computing the following cosine distance:

$$M_{ij} = \text{CosineDist}(h^j(X), \tilde{h}_i^j(X)).$$

That is, $M_{ij}$ quantifies the dependency of block $j$ on block $i$. Thus, a small cosine distance in-

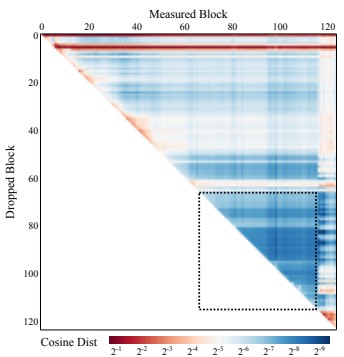

Figure 3: Block-wise dependency heatmap of Ultra-253B-Base before applying FFN Fusion (log-scale). Each coordinate $(i, j)$ encodes how much block $j$ depends on block $i$, measured by the cosine distance between $h^j(X)$ and $\tilde{h}_i^j(X)$.

dicates that dropping block $i$ has little effect on block $j$, suggesting relative independence—a characteristic that can be exploited for increasing parallel computation. Conversely, a large cosine distance corresponds to a strong dependency, implying that sequential processing is more critical to maintain performance. We illustrate this approach in Figure 3 by constructing a dependency matrix, $M$, for Ultra-253B-Base (prior to FFN Fusion; see Section 4). This matrix visually encodes the interdependencies among the model's layers: darker blue hues indicate weaker dependencies—signaling promising opportunities for FFN Fusion—while darker red hues denote strong dependencies that would hinder parallelization or fusion. For example, Figure 3 shows that all layers in the model depend on layers 0 and 5, causing their entire rows to be colored in dark red hues. The dark blue region, marked with a dashed square, shows a sequence of FFNs with low interdependency were selected for FFN Fusion. Due to GPU memory constraints, we are unable to fuse them all at once, so we divide them into fusion sequences based on the maximum size that fits within our devices. Additionally, we applied this metric to explore *block parallelization* (Apendix B), aiming to parallelize general LLM blocks rather than just FFNs. We constructed the dataset that was used for this evaluation following *Distillation Mix* [3].

## 4 Producing Large-Scale Models with FFN Fusion

In this section, we describe how FFN Fusion is applied at large scale to transform a 405B-parameter Llama-based model into our more compact Ultra-253B-Base model. Specifically, we show how identifying and fusing long sequential feed-forward blocks reduces depth without sacrificing accuracy. We utilize a lightweight refinement KD and alignment phase to ensure the fused model retains or even improves upon the performance of its larger predecessor. Finally, we present a comprehensive evaluation of the resulting model, demonstrating significant speedups and strong results on standard benchmarks. We first run the standard Puzzle search on Llama-405B, specifying that the derivative model must obtain a $1.5\times$ latency speedup and fit within a single NVIDIA $8\times$H100 node (640 GB total), and in a single B100 GPU (192 GB) . This yields a 253B-parameter model whose overall configuration is shown in Appendix D. Many attention layers were removed, resulting in 50 blocks contiguously arranged without interleaved attention layers.

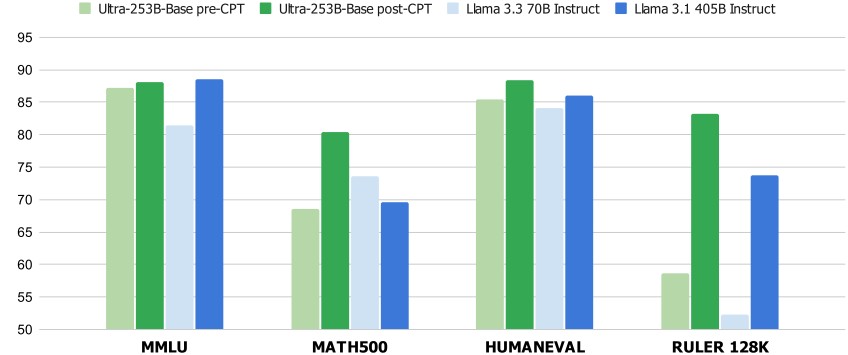

Figure 4: Comparison of Ultra-253B-Base before and after applying an additional CPT.

**FFN Fusion.** Next, we apply FFN Fusion (§3) to 49 of the 50 consecutive FFN layers (See Section 5.3 for ablation on leaving the last layer). Due to per-GPU memory limits, we split the layers into four sequences [66, 73], [74, 85], [86, 100], [101, 114], each fused into a single FFN. Before fusion, the baseline model achieved 84.23 on MMLU and 8.83 on MT-Bench. Remarkably, after fusing all 49 layers—layers that, if *removed* entirely, would damage performance severely—the model still maintains similar MMLU and MT-Bench scores of 82.76 and 8.35 before any additional training.

**Additional Training.** To recover performance, we used KD as described in [3]. This involved a multi-stage distillation process: 54B tokens at 8k context, followed by 5B tokens each at 16k and 32k, and finally 0.8B tokens at 128k. The KD process improved MMLU and MT-bench scores to 85.17 and 9.10, respectively. Further optimization was explored through two methods. First, inexpensive alignment via RLHF [45]. Table 1 demonstrated that Ultra-253B-Base surpassed Llama-405B's capabilities, par-

Table 1: Comparison of Ultra-253B-Base and its parent Llama-405B after applying FFN Fusion, KD, and alignment.

| Metric | Llama-405B | Ultra-253B-Base |
|---|---|---|
| MMLU Instruct [32] | 86.53 | 87.54 |
| MMLU-Pro [44] | 71.92 | 72.25 |
| Arena Hard [29] | 72.56 | 84.92 |
| HumanEval [4] | 85.97 | 86.58 |
| MT-Bench [50] | 9.06 | 9.19 |

ticularly on the Arena Hard benchmark. We applied only a longer continual pretraining (CPT), without alignment, following the initial KD. This involved 73B tokens at a context length of 8k and another 15B tokens at a context length of 258k, and also yielded strong performance, even before instruction-based tuning (Figure 4). Even after removing nearly half of the model's attention layers, we still improved its long-context performance, outperforming Llama-405B on RULER-128K [22].

**Efficiency Improvements.** Ultra-253B-Base achieves a 1.71× speedup in user latency (Table 2), a 35× lower per-token cost at batch size 32, and reduced memory footprint with half the attention layers and 253B parameters (down from 405B). Ultra-253B-Base breaks the efficient frontier on a single H100 node, offering a state-of-the-art accuracy and latency under the similar hardware constraints (see Figure 5). Table 2 details the user latency (tokens/second) achieved by Llama-405B, by 253B model (with and without FFN Fusion), and by Llama-3.3-70B, all under identical tensor

Table 2: User latency under Tensor Parallel (TP) 8 on a single H100 node. A higher tokens/second value indicates lower latency for a single user.

| Model | Tokens/Second |
|---|---|
| Llama-405B | 41.44 |
| 253B PreFusion | 63.38 (1.53×) |
| **Ultra-253B-Base** | 70.92 (1.71×) |
| Llama-3.3-70B | 94.03 |

parallel settings on a single 8×H100 node. Notably, Ultra-253B-Base is 1.71× faster than the parent for single-user decoding. On NVIDIA H200, its rate increases to 90.05 tokens/second. With speculative decoding [27]—using Llama-3.2-1B-Instruct as a draft model with no extra training—Ultra-253B-Base reaches 202 **tokens/s** on H200. This speculative decoding latency is averaged over all MT-Bench completions. Notably, as TP size increases, the efficiency of the fused FFN layers will increase further, and thus ready to benefit from improving hardware designs, such as GB200 NVL72 nodes. Overall, Ultra-253B-Base demonstrates that FFN Fusion coupled with the Puzzle algorithm can greatly reduce a model's depth at large scale.

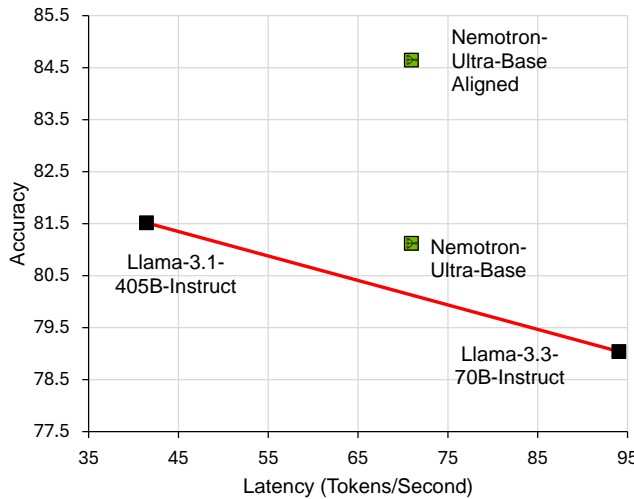

Figure 5: Accuracy vs. latency performance of Ultra-253B-Base. Latency is measured on a single NVIDIA H100 node with tensor parallel (TP) 8, running in FP8. The red line represents the efficient frontier, highlighting models with the best accuracy-to-throughput tradeoff. Accuracy = $(\text{MT-Bench} \times 10 + \text{MMLU} + \text{MMLU-Pro} + \text{Arena Hard} + \text{HumanEval})/5$.

## 5 Additional Empirical Studies

In this section, we present empirical study to validate our approach. In Section 5.1 we present results for FFN Fusion on a 70B model. Next, in Section 5.2, we examine the consequences of removing FFN layers rather than fusing them, demonstrating that these layers are vital to preserving model quality. In Section 5.3, we investigate the sensitivity of certain FFNs to fusion. Section 5.4 demonstrates that the FFN Fusion phenomenon is applicable to a variety of models, spanning different scales and families. Finally, in Section 5.5 we provide a hypothesis as to why FFN Fusion works.

### 5.1 FFN Fusion in 70B scale

We consider a derivative of Llama-3.1-70B-Instruct created using the Puzzle algorithm [3], which reduces the model to 49B parameters while matching the original accuracy. The Puzzle process prunes attention in specific layers, leaving two main sequences of consecutive FFN layers: layers 42–51 (10 layers) and layers 53–70 (18 layers). We evaluate FFN Fusion at four progressively increasing levels of intensity, each reducing the number of FFN layers more than the previous:

Table 3: Evaluation of FFN Fusion. The Fusion column indicates how many FFN layers were replaced by the corresponding number of new layers.

| Model | Fusion | MMLU | MT-Bench | Accuracy |
|---|---|---|---|---|
| Baseline | - | 80.73 | 8.87 | 84.71 |
| Step 1 | $26 \rightarrow 12$ | 80.64 | 8.72 | 83.92 |
| Step 2 | $26 \rightarrow 6$ | 80.29 | 8.54 | 82.84 |
| Step 3 | $26 \rightarrow 3$ | 80.39 | 8.30 | 81.69 |
| Step 4 | $26 \rightarrow 2$ | 79.98 | 8.25 | 81.24 |

- **Step** 1: Fuse adjacent pairs of FFNs within each sequence of consecutive FFNs.
- **Step** 2: Merge neighboring pairs from Step 1 to form longer fused blocks of length 4 or 5.
- **Step** 3: Fuse the entire first sequence (layers 42–50) into a single FFN, and split the second sequence (53–69) into two fused blocks.
- **Step** 4: Fuse the entire second sequence (53–69) as well, reducing each main FFN run to a single layer.

Similarly to the Ultra-253B-Base, we chose to exclude the last FFN in each sequence. We perform no additional training when applying these fusions. Table 3 reports each variant's performance on MMLU and MT-Bench, as well as a combined score Accuracy = $(\text{MMLU} + 10 \times \text{MT-Bench})/2$. We observe that step-1 fusion reduces depth with only a 1% overall accuracy drop, while step-4 fusion (collapsing each sequence to a single layer) sees roughly a 4% drop.

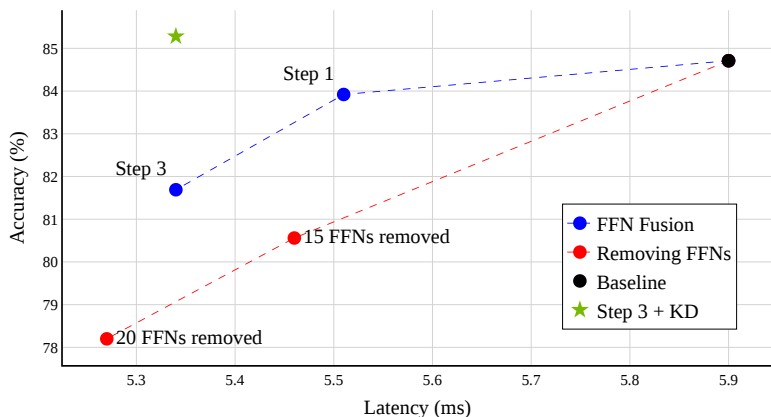

Figure 6: Accuracy vs. Latency for FFN Removal vs. Fusion.

## 5.2 Removing FFNs vs. FFN Fusion

An intuitive alternative to fusing FFNs is to *remove* the same FFN layers outright. However, as shown below, large-scale removal generally leads to significant accuracy degradation, whereas fusion preserves representational capacity by retaining all parameters in a single, parallel module. To determine which FFNs to drop, we rely on Puzzle's block-importance scores and remove the least important FFNs first. We restrict removal only to the FFN-fusion regions. For the 49B model, we observe a clear advantage for fusion over removal. Figure 6 compares the accuracy-latency trade-off for two removal levels (15 or 20 FFNs removed) versus the two fusion steps. As we gradually remove more FFNs, we decrease latency but see larger accuracy drops. By contrast, fusing step 3 yields a 10% latency improvement with only a 3.5% accuracy drop. Removing 15 or 20 FFNs yields comparable or slightly higher latency gains (7.5% or 11%), but at steeper accuracy declines (5% or 8.7%). Moreover, a brief knowledge-distillation phase (25B tokens) on the step-3 fused model actually surpasses the baseline (MMLU: 80.56, MT-Bench: 9.00).

## 5.3 The Final FFN in Each Sequence is Sensitive to Fusion

With the 49B model, we observe that fusing the final FFN in a long sequence often degrades accuracy more than fusing earlier FFNs. Below, we summarize an ablation study to highlight this phenomenon. Table 4 examines various ways of fusing the two main FFN sequences (layers 42–51 and 53–70) in the 49B model. The first part of the table compares fusing the entire second sequence ([53, 69] vs. [54, 70]) and the entire first sequence ([42, 50] vs. [43, 51]) in isolation, indicating that layer 70 is especially sensitive. In the second part, we fuse both sequences simultaneously. When either layer 51 or layer 70 is included in these large fused blocks, we see additional performance drops. Notably, skipping the final FFN (e.g. fusing [42, 50] , [53, 69]) yields higher MT-Bench scores than including layer 51 or 70. The final FFN in each attention-removed sequence appears uniquely important to the model's representations.

Table 4: Evaluating the impact of fusing the final FFN in 49B scale. Fusing these final layers often causes more accuracy loss.

| Sequence | MMLU | MT-Bench |
|---|---|---|
| No Fusion | 80.73 | 8.87 |
| [53, 69] | 80.57 | 8.49 |
| [54, 70] | 79.89 | 8.12 |
| [42, 50] | 80.49 | 8.39 |
| [43, 51] | 80.46 | 8.42 |
| [42, 50], [53, 69] | 79.98 | 8.25 |
| [42, 51], [53, 69] | 80.05 | 7.64 |
| [43, 51], [54, 70] | 79.92 | 7.38 |
| [42, 51], [53, 70] | 79.89 | 7.30 |

While most layers can be safely fused, incorporating the last FFN often triggers a significant accuracy drop. Consequently, omitting this final FFN from the fused groups is typically a more reliable choice for efficient fusion with minimal performance loss.

## 5.4 Additional Models

This section explores FFN Fusion on additional models, evaluating its effectiveness across different scales and model families. We applied FFN Fusion to Llama-3.1-8B-Instruct [13] and Mistral Large 2 following a similar procedure as with our 253B and 49B models. Specifically, we used the Puzzle algorithm on both models to remove a portion of their attention layers. Following this, we fused consecutive sequences of FFNs. For Mistral Large 2, we removed 27 attention layers that resulted in 2 sequences of 14 and 13 consecutive FFN layers. We fused each sequence (excluding the last FFN) to a single FFN layer. For Llama-3.1-8B-Instruct (32 layers), we removed 8 consecutive attention layers. We fused the sequence (excluding the last FFN) to a single FFN layer. We also conducted a short KD training on the 8B model over 20B tokens and compared the results to the original model. Table 5 provides a clear demonstration of FFN Fusion's generalizability, showing its effectiveness across additional models.

Table 5: FFN Fusion impact on Mistral Large 2 and Llama 3.1 8B Instruct.

| Fusion | MMLU | MT-Bench |
|---|---|---|
| Mistral Large 2 | | |
| ✗ | 80.56 | 7.16 |
| ✓ | 80.06 | 6.76 |
| Llama 3.1 8B Instruct | | |
| ✗ | 68.91 | 8.28 |
| ✓ | 68.22 | 7.99 |

## 5.5 Fusion Explainability

In this section, we aim to further explain the phenomenon of FFN Fusion. We examine the functional structure of LLMs that allows for the fusing of layers, specifically focusing on the consecutive FFN layers after attention removal. First, we rewrite the token-level normalization equation in an alternative form: $\eta_k(x_i) = D_{s_k} x_i / \|x_i\|$, where we replace the normalization parameters $s_k \in \mathbb{R}^d$ with a diagonal matrix $D_{s_k} \in \mathbb{R}^{d \times d}$, with $s_k$ on the diagonal. This reformulation allows us to "push" the matrix into its corresponding module—either attention or FFN—and

Table 6: Reverse order vs Fusion experiment on Puzzle-49B.

| Sequence | Fusion Accuracy | Reverse Accuracy |
|---|---|---|
| [42, 49] | 83.74 | 83.52 |
| [53, 60] | 83.24 | 83.25 |
| [63, 67] | 84.22 | 84.10 |
| [63, 70] | 84.00 | 83.88 |

consider equations 1 and 4 as functions of the token direction $\overrightarrow{x_i} = x_i / \|x_i\|$, while ignoring the magnitude. Figure 11 (a and b), Similar to the measurements in [30], shows the cosine distance between $\boldsymbol{X}$ and $f(\boldsymbol{X})$ for each layer of both models. The plot reveals that the FFN fusing areas—$[42, 51]$, $[53, 70]$ for the 49B model and $[66, 115]$ for the 253B model—exhibit lower cosine distance compared to other regions in the model. Assuming that small changes in the input to an FFN layer lead to small changes in its output, we can conclude that fusing the FFNs with low cosine distance (except for the last one) will not cause significant changes to the model. Specifically, by fusing the FFN layers, we are altering the input for each layer, but the directional difference between the original input (according to the original model's structure) and the new input (resulting from fusion) remains small, and the outputs should remain similar under the smoothness assumption. Another experiment (Table 6) that strengthens this claim shows that even if we reverse the FFNs order instead of fusing them, we get similar performances. The results in Figure 11 (c and d) present the relationship between the layer input $\boldsymbol{X}$ and $h(\boldsymbol{X}) = f(\boldsymbol{X}) - \boldsymbol{X}$. The small change in token directions may be linked to the low cosine distance between $h(\boldsymbol{X})$ and $\boldsymbol{X}$, but they are nearly orthogonal (with a distance of around 0.95 for both models across all layers). However, a more suitable explanation comes from the ratio $\|h(\boldsymbol{X})\| / \|\boldsymbol{X}\|$, which is smaller in the fused regions. This suggests that $h(\boldsymbol{X})$ is small compared to $\boldsymbol{X}$ that it does not significantly affect the direction of $\boldsymbol{X}$. Although this section can also serves as motivation for FFN removal, we demonstrated in Figure 6 that removing these layers significantly harms the model's performance, to a much greater extent than fusion does. Both metrics were evaluated using the *Distillation Mix* dataset.

# 6 Concluding Remarks

This work introduces FFN Fusion, a novel optimization technique that demonstrates how sequential computation in large language models can be effectively parallelized. Through comprehensive experiments at scales from 49B to 253B parameters, we showed that sequences of FFN layers can be fused with minimal impact on model capabilities. Our work revealed couple of key insights: (1) FFN layers in attention-removed regions exhibit remarkably low inter-layer dependencies, suggesting these

computations may be more independent than the sequential architecture implies; and (2) empirically, final FFN layers in parallelizable sequences show higher sensitivity to fusion—an observation that points to potentially meaningful structural patterns in how these models process information.

The practical impact of our work is demonstrated through Ultra-253B-Base, which achieves a $1.71\times$ speedup in inference latency compared to its parent Llama-405B model, while requiring $35\times$ lower per-token cost at batch size 32. Most remarkably, these efficiency gains come with minimal degradation in model capability—in fact, Ultra-253B-Base matches or exceeds its parent's performance across key benchmarks despite using only 253B parameters compared to the original 405B.

Our findings open several promising research directions. First, the clear patterns in inter-layer dependencies we observed could provide a new lens for model interpretability research. The ability to identify which FFN layers operate independently versus those that require sequential processing may offer insights into how these models structure and transform their internal representations. Second, our preliminary finding that even full transformer blocks can sometimes be parallelized suggests possibilities for new architectural designs explicitly optimized for parallel execution. Third, future work could extend FFN Fusion to models incorporating Mixture-of-Experts (MoE) layers. Finding a way to fuse MoE layers while maintaining sparse activation patterns and efficient expert routing could lead to efficiency improvements. Finally, the strong performance of FFN Fusion at larger scales raises intriguing questions about the relationship between model size and natural parallelization.

## 7 Societal Impact

The inference efficiency achieved through FFN Fusion, as demonstrated by its impact on operational costs in Section 4, significantly improves the accessibility of large language models. By reducing these costs, a wider array of users and organizations can now better afford to deploy and leverage these powerful technologies, effectively dismantling economic barriers.

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

## A   Proof of Theorem 3.1

*Proof.* The proof will proceed by induction. For the base case, we consider two FFN layers, $\text{FFN}^1$ and $\text{FFN}^2$. We will demonstrate that $\text{FFN}^* = \text{FFN}^1 + \text{FFN}^2$. The induction step follows naturally from the associative property of addition.

$$
\begin{aligned}
(\text{FFN}^1(\boldsymbol{X}) + \text{FFN}^2(\boldsymbol{X})) =& (\sigma(\boldsymbol{X}(W_2^1)^T) \odot \boldsymbol{X}(W_1^1)^T)(W_3^1)^T + (\sigma(\boldsymbol{X}(W_2^2)^T) \odot \boldsymbol{X}(W_1^2)^T)(W_3^2)^T \\
=& \left[ \sigma(\boldsymbol{X}(W_2^1)^T) \odot \boldsymbol{X}(W_1^1)^T, \sigma(\boldsymbol{X}(W_2^2)^T) \odot \boldsymbol{X}(W_1^2)^T \right] (W_3^*)^T \\
=& (\left[ \sigma(\boldsymbol{X}(W_2^1)^T), \sigma(\boldsymbol{X}(W_2^2)^T) \right] \odot \left[ \boldsymbol{X}(W_1^1)^T, \boldsymbol{X}(W_1^2)^T \right])(W_3^*)^T \\
=& (\sigma(\boldsymbol{X}(W_2^*)^T) \odot \boldsymbol{X}(W_1^*)^T)(W_3^*)^T \\
=& \text{FFN}^*(\boldsymbol{X})
\end{aligned}
$$

where $[\cdot, \cdot]$ is the concatenation of matrices along the second dimension. $\qquad\square$

## B   Block Parallelization

In all previous experiments, we focused on fusing sequences of FFNs in attention-removed layers (see Section 3). In that setting, the homogeneous structure of FFN-only layers permits straightforward weight concatenation and fusion. However, when considering full block parallelization—where entire Transformer blocks, each comprising both an attention module and an FFN, are run in parallel—such fusion is not possible. This is because each full block consists of two distinct components, with the attention output explicitly directed to its paired FFN. Running entire blocks completely in parallel is not currently natively supported by heavily optimized inference frameworks such as TensorRT-LLM or vLLM [24]. Nevertheless, one can envision assigning each full block to a different GPU, thereby maximizing parallelism and potentially achieving significant speedups. We use our block-wise dependency analysis to examine the dependency structure between blocks and identify candidate groups whose outputs are relatively independent. Such groups are ideal for parallelization with minimal accuracy loss and could enable higher degrees of tensor parallelism—each full block operating on its own set of GPUs—thereby improving inference throughput. In our experiments, we implement these full block parallelization strategies using a more flexible environment (e.g., HuggingFace Transformers [46]), albeit with reduced optimization for large-scale deployments. A thorough investigation of the actual speedups in specialized frameworks is left for future work.

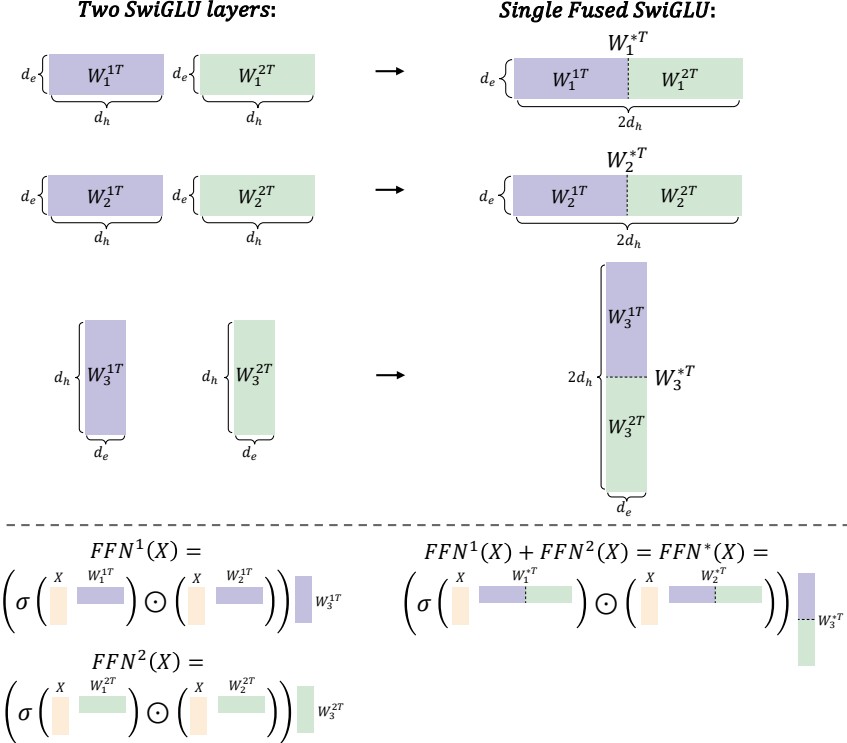

Figure 7: A visualization of FFN Fusion applied to SwiGLU. Two FFNs (left) are fused into a single FFN (right).

**Method.** We first compute the *block-wise dependency matrix* exactly as in Section 3, We then search for *candidate sequences* of 4 consecutive blocks to fuse. The choice of this number of blocks is mostly related to hardware constraints. Arbitrary sizes will be considered in our future work. For each such sequence $[i, i+3]$, $i \in [m-4]$, we extract the corresponding $\boldsymbol{M}^{[i,i+3]} \in \mathbb{R}^{4 \times 4}$ submatrix from the block dependency matrix and compute two statistics:

$$\boldsymbol{M}^{[i,i+3]}_{\max} = \max_{k,j \in [4]} \boldsymbol{M}^{[i,i+3]}_{kj}, \quad \boldsymbol{M}^{[i,i+3]}_{\text{sum}} = \sum_{k,j \in [4]} \boldsymbol{M}^{[i,i+3]}_{kj}$$

A lower $\boldsymbol{M}_{\max}$, and $\boldsymbol{M}_{\text{sum}}$ indicates weaker dependencies among those blocks, suggesting they may be more amenable to parallelization. We exclude any blocks that lack attention (shaded regions in Figure 8b) because our focus here is on *full* Transformer blocks that contain both attention and FFN modules. We employ a simple *greedy algorithm* to choose the best subsequences:

1. Choose the length-4 sequence starting index $i^* = \arg\min_i \boldsymbol{M}^{[i,i+3]}_{\max}$ .

2. If the argmin is not unique, break the tie using $i^* = \arg\min_i \boldsymbol{M}^{[i,i+3]}_{\text{sum}}$ (restricted to the set from step 1).

3. Parallelize the blocks according to the chose sequence $[i^*, i^* + 3]$.

4. Remove any overlapping sequences.

5. Repeat until no valid sequences remain.

The motivation behind this algorithm is to avoid choosing sequences with high dependency pairs. The sum tie breaking is due to step-like characteristic of $\boldsymbol{M}_{\max}$ statistic (see Figure 8b). Table 7 reports the downstream performance (MMLU and MT-Bench) after fusing the sequences chosen by 4 algorithm steps.

**Results.** From Table 7, we observe that parallelizing the first and second sequence of four blocks leads to only a modest drop in MMLU, but once the third sequence is added, the decline becomes much more pronounced. This suggests that full block parallelization is more challenging than FFN Fusion. These observations are supported by the block-wise dependency heatmap and the submatrix statistics (Figure 8), both of which indicate stronger inter-block dependencies and higher $M_{max}$ and $M_{sum}$ values in full block sequences compared to attention removed sequences. In addition, Figure 11 shows that the full transformer blocks alter significantly the tokens directions.

Table 7: Comparison of MMLU and MT-Bench scores across different parallelization strategies .

| Model Name | MMLU | MT-Bench |
|---|---|---|
| 49B model (no fusion) | 80.73 | 8.87 |
| **Parallel Sequences** | | |
| $[38, 41]$ | 80.51 | 8.67 |
| $[38, 41], [71, 74]$ | 80.04 | 8.67 |
| $[38, 41], [71, 74], [32, 35]$ | 77.86 | 8.20 |
| $[38, 41], [71, 74], [32, 35], [26, 29]$ | 56.12 | 6.62 |

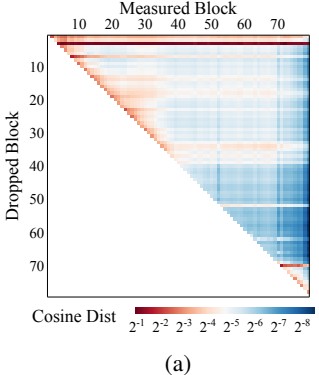

(a)

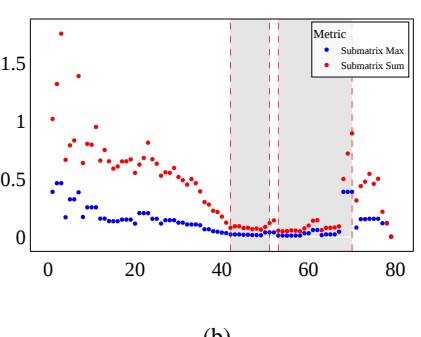

(b)

Figure 8: (a) Block-wise Dependency Heatmap of the 49B model (log-scale). Darker blue hues indicate weaker dependencies, darker red hues indicate stronger dependencies. (b) $M_{max}$ and $M_{sum}$ values for 4-Block Sequences of the 49B model. Lower values indicating more promising candidates for parallelization.

## C  Datasets

For applying Puzzle throughout our experiments, we used the same dataset mixture used in [3], termed *Distillation Mix*. This mixture includes source code repositories, Wikipedia articles, books, news websites, and several other domains. The dataset comprises 224 billion tokens collected from three public datasets: FineWeb [34], Dolma [40], and Buzz-V1.2 [21]. For Ultra-253B-Base KD training we used the same data reinforced with synthetic data generated with Llama-405B, following the [48] approach, to help further align Ultra-253B-Base with its parent model.

## D  Ultra-253B-Base and 49B Architecture Overview

In this section, we detail the configuration of each block in the Ultra-253B-Base model and highlight several unique design choices introduced by the Puzzle framework on top of Llama-405B. Notably, the model employs variable FFN widths, with scaling multipliers ranging from $0.4875$ up to $4.875$ across its 126 blocks. This variability allows the model to dynamically adjust capacity at different depths, balancing performance with computational efficiency. Furthermore, it includes a sequence of 50 consecutive layers that bypass the attention mechanism (denoted by `no_op`), dedicating these layers entirely to FFN processing and creating an ideal region for FFN Fusion.

We observe a similar pruning pattern in the 49B model, which is derived from the Llama-3.1 − 70B model using the Puzzle framework. Like its larger counterpart, the 49B model also features variable FFN multipliers and blocks where attention is removed, although at a smaller scale. Figure 9 compares both architectures.

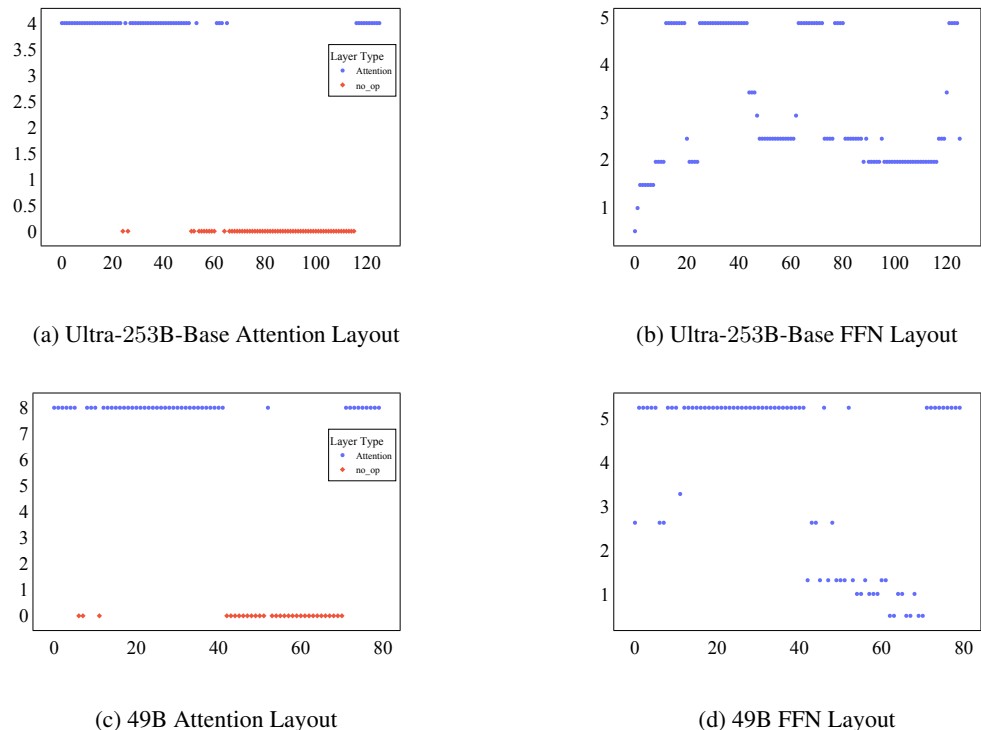

(a) Ultra-253B-Base Attention Layout

(b) Ultra-253B-Base FFN Layout

(c) 49B Attention Layout

(d) 49B FFN Layout

Figure 9: A $2 \times 2$ overview of Ultra-253B-Base (top row) and 49B model (bottom row). Subfigures (a) and (b) illustrate the attention and FFN configurations, respectively, for the 253B model. Subfigures (c) and (d) show the corresponding layouts for the 49B model. Both architectures feature variable FFN widths and regions where attention has been removed, although at different scales.

# E  MoE Inference Performance

MoEs show great promise in reducing inference costs and are currently spearheaded by DeepSeek-V3 [6], a highly sparse MoE with a great performance. However, MoEs have their own problems in inference.

## E.1  Problems with Small Blocks

An underlying problem for MoEs paradoxically arises from the small size of their sub-modules. Smaller layers incur large latency than expected for two reasons:

- GPU utilization - low level GPU overheads such as wave quantization produce latency problems that become stronger as the module becomes smaller.

- Communication overhead - The All-Reduce operation that is typical for tensor parallelism creates an overhead for communication. For small modules this overhead increases in proportion.

MoE's experts are smaller than dense MLPs and hence suffer more strongly from the above effects. In addition the routing mechanism of MoEs is another small module that suffers accordingly. As a result, dense models scale better with the number of GPUs while MoEs reach an early saturation. It is worth to notice that this is in complete contrast to our FFN Fusing method that creates larger modules that parallelize better, as visualized in Figure 2.

## E.2 Batch Size Effects

MoEs behave very well for very large batch sizes. For such large workloads load balancing issues are less dominant and low level overheads become negligible. However, smaller batch sizes are not as efficient. Concretely, we define an efficient size of a batch through the perspective of analyzing a linear layer. The performance of linear layers varies significantly with batch size. For small batches, performance is limited by the time needed to read weights from DRAM (I/O bottleneck). For large batches, performance becomes compute-bound, limited by the hardware's maximal FLOPs/sec. Notably, the latency of a linear layer remains the same for batch sizes $1$ and $64$, as both are dominated by weight loading time. This makes small batches extremely inefficient. While other factors exist, we consider a linear layer to operate efficiently once the number of tokens in the batch approaches $64$, and inefficient otherwise. MoEs require all experts to be fully utilized for high throughput. in practice, the total batch is split across experts, significantly reducing the effective number of tokens each expert sees. For example, with a sparsity factor of $1/32$ (as in [5, 6]), a batch of $2048$ tokens ($64 \times 32$) is needed to ensure each expert processes around $64$ tokens. The case is more extreme in Llama $4$ Maverick, where a $1/128$ sparsity factor requires a batch of $8192$ tokens ($64 \times 128$) to reach this threshold. These batch sizes are often impractical—especially in generation mode—leaving MoEs to operate in a low-efficiency regime, where linear layers are under-utilized. We can thus conclude that MoEs only really live up to their promise for very large batch sizes. For more commonly used intermediate batch sizes they suffer from bad scaling with the number of tokens, larger low level overheads, and worse parallelization scaling than dense models.

## F Further Pairwise Block Dependency Analysis

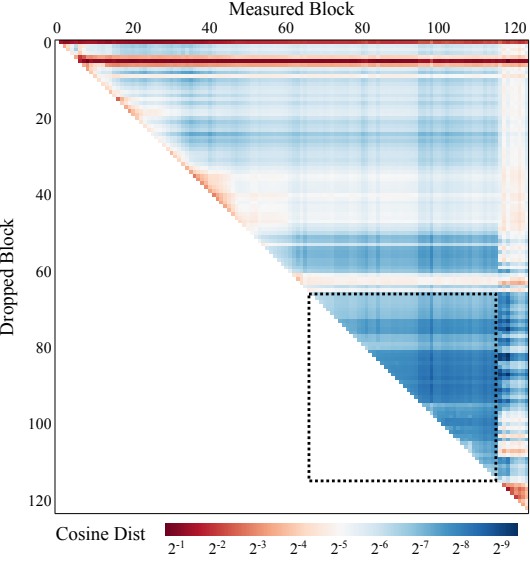

Figure 10: Replication of the block-wise dependency heatmap for Ultra-253B-Base PreFusion, shown here for convenience. Each coordinate $(i, j)$ represents the cosine distance between $h^j(\boldsymbol{X})$ and $\tilde{h}_i^j(\boldsymbol{X})$, quantifying how much block $j$ depends on block $i$. *Darker blue* indicates weaker dependencies, while *darker red* indicates stronger dependencies. The dotted box marks an attention-removed region with generally low dependency values, suggesting high parallelization potential.

Figure 10 offers a more detailed view of the block-wise dependency structure in Ultra-253B-Base PreFusion. Below, we highlight several notable observations:

- **Overall Color Scale.** Each entry $(i, j)$ in the heatmap corresponds to the cosine distance between the outputs of block $j$ when block $i$ is dropped versus when the model is intact. A *dark red* hue signifies a strong dependency, meaning that omitting block $i$ substantially alters block $j$. Conversely, *dark blue* indicates a weak dependency, suggesting that dropping block $i$ does not notably affect block $j$.

- **Attention-Removed Region (Dotted Box).** The dotted box highlights a region of consistently low dependency values, where most blocks exhibit *minimal* mutual influence. This area arises from attention pruning in those layers and is particularly well-suited for FFN Fusion, since layers here can be more easily parallelized without significantly harming accuracy.

- **Crucial Early Blocks.** Some blocks in the upper rows of the matrix (toward the top of the y-axis) display consistently *red* cells across many columns. This indicates that certain early layers have a strong influence on a wide range of subsequent blocks, making them less amenable to parallelization.

- **Sub-region with Sequential Reliance.** A noticeable diagonal band of higher values appears in some sub-regions, implying that each block in that band strongly depends on its immediate predecessor. Such sequential reliance reduces the potential for parallelization in those areas, since omitting any single block significantly affects the next one.

- **Global Sinks at Later Layers.** As we move toward deeper layers (near the bottom-right corner), some blocks again show stronger dependencies, acting as "global sinks" that consolidate information from many earlier blocks. Although the importance of certain blocks may wane in mid-network regions, it can resurface in the final layers.

In summary, the dependency matrix reveals both strongly and weakly coupled sub-regions. High-dependency zones require careful consideration to avoid accuracy loss if parallelization is attempted. Conversely, low-dependency zones, such as the attention-removed region, offer a promising avenue for efficient FFN Fusion.

## G   Fusion Explainability

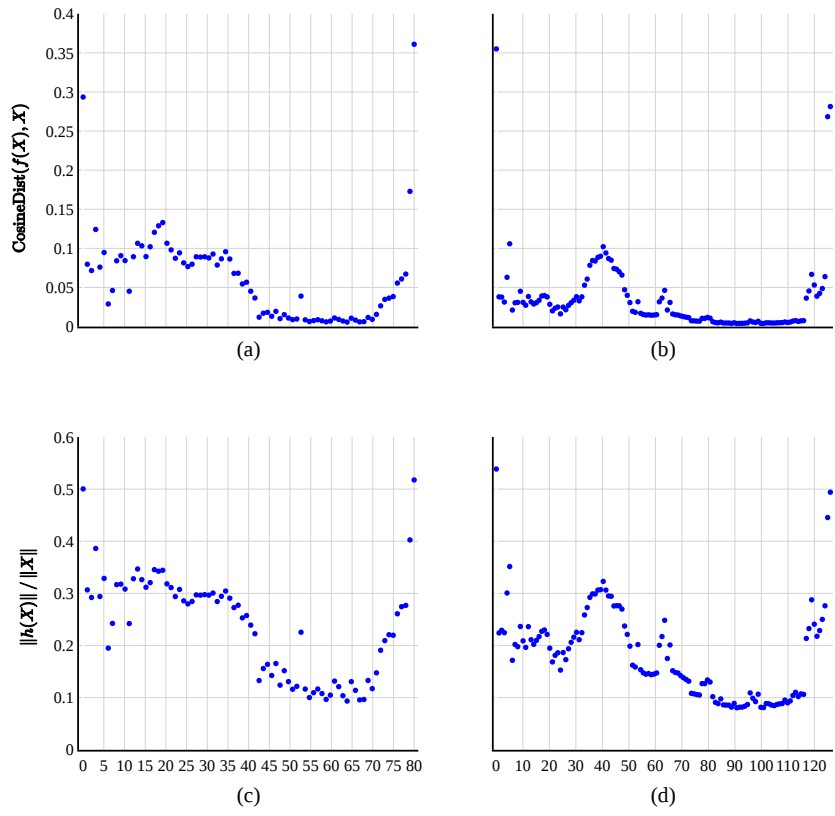

Figure 11: Per-layer metrics. Upper row is the cosine distance between $f(X)$ and $X$ for the (a) The 49B model and (b) Ultra-253B-Base model. Bottom row represents the ratio between $h(X)$ and $X$ for the (c) The 49B model and (d) Ultra-253B-Base model.

