# OpenReview forum: "FFN Fusion: Rethinking Sequential Computation in Large Language Models"
_NeurIPS.cc/2025/Conference — NeurIPS 2025 spotlight_

### Official Review · Reviewer_Cc25 · 2025-06-17

**Clarity:** 2
**Significance:** 2
**Originality:** 3
**Rating:** 4
**Confidence:** 3

**Summary:**

The paper proposes a modification to the existing LLM architecture by: 1) removing some attention layers (using an existing method called Puzzle), and 2) fusing sequential FFNs to improve the runtime efficiency of LLMs. The authors apply these techniques to Llama-3.1-405B-Instruct and create a 253B model, referred to as 253B-Fusion, which achieves a 1.71× inference speedup while maintaining accuracy. Knowledge distillation is used to recover model quality.

**Questions:**

1. The majority of the performance gains come from the existing Puzzle method rather than Fusion.
To construct the fused model, the authors first derive a pre-fusion 253B model by applying Puzzle to Llama-405B. From Table 2, this model already achieves a 1.53× speedup over the original 405B model, primarily due to reduced model size and the elimination of attention layers. The current work proposes further speedup by fusing sequential FFNs into a wider FFN to enhance GPU utilization. However, this yields only about a 12% additional speedup, which is relatively minor considering it is a lossy transformation.

2. The speedup from fusing large matrix multiplications offers limited potential.
In large models, the FFN layers themselves are quite large. As GPU memory capacity increases, larger weight slices can be stored on each GPU under tensor parallelism. This reduces the benefits of fusion, thereby limiting its potential for further speedup.

**Ethical Concerns:**

["NO or VERY MINOR ethics concerns only"]

**Final Justification:**

The response of the authors address some of my concerns (the second weakness) but does not address the first weakness in my review. But I am in general positive for this paper. I tend to keep my weak accept score.

**Quality:**

3

**Strengths And Weaknesses:**

**Strengths**
1. The topic of exploring runtime-efficient LLM architecture is important.
2. The authors provide solid experimental results validating the fused model’s accuracy.

**Weaknesses**
1. The majority of the performance gains come from the existing Puzzle method rather than the proposed Fusion technique.
2. The speedup from fusing large matrix multiplications offers limited potential.

---

> ### Author Rebuttal · Authors · 2025-07-30
>
> Thank you for your positive feedback,
>
> *”The speedup from fusing large matrix multiplications offers limited potential. In large models, the FFN layers themselves are quite large. As GPU memory capacity increases, larger weight slices can be stored on each GPU under tensor parallelism. This reduces the benefits of fusion, thereby limiting its potential for further speedup.”*
>
> FFN Fusion delivers *larger* speed‑ups on newer and better GPUs. To empirically demonstrate this, we compared the original Llama‑405B and our 253B‑Fusion on TP 8 B200 nodes, observing a 1.83× latency improvement, which exceeds the 1.71× measured on TP 8 H100.
>
> If we understand your argument correctly, you suggest that increased GPU memory in newer GPUs would allow running with lower TP (otherwise, with the same TP the same-sized weight slices would be stored on each GPU), but this is not necessarily true.
> A complete cost model must also weigh communication. **Latency**‑sensitive inference typically runs at small batch sizes where high‑bandwidth memory (HBM), not total memory DRAM, is the bottleneck. Even when the entire model fits on one newer GPU—for example, 253B‑Fusion at FP4 on a single B100 or B200—running in TP to shorten the critical path of reading weights from HBM and to share the workload across multiple devices is still preferable. At these small batch sizes the I/O of fetching weights, not raw compute, limits latency, so only greater HBM bandwidth or distributing the read over several GPUs helps; simply adding more DRAM capacity does not. Using TP, however, inserts a synchronization for every layer, which can be eliminated using FFN Fusion. Thus the relative advantage of FFN Fusion increases as TP deepens and as precision is reduced, because both trends amplify communication costs.
>
> To illustrate this point, we benchmarked 253B-Fusion with FP4 weights on a B200 GPU, and compare it to the 253B-Fusion measurements for FP8 in the manuscript. With no TP, a single B200 run is 0.53× slower than TP 8 on H100, highlighting the HBM’s bandwidth ceiling. Moving to TP 2 on B200 closes the gap to 0.98×, and at TP 8 the same model runs 2.8× faster than TP 8 on H100.
>
> Furthermore, FFN Fusion is designed to improve practical aspects of performance that arise from inefficient matrix multiplications (GEMMs) for matrices that are too small compared to the GPU capabilities (See Appendix E.1.), and to alleviate inter-device communication overheads by reducing the number of layers.
> Each FFN matrix looks “large” on paper, yet under TP it is sliced into tiles that may occupy only a fraction of a GPU’s streaming multiprocessors (SMs). Let us consider a practical example. In Llama 70B each MLP weight matrix is 8K*28K or 28K*8K (224MB in total assuming FP8 data type). Running with TP8 each GPU weight matrix is sharded to just 8K*3.5K or 3.5K*8K (224MB/8=28MB). The sizes (and number of tiles) discussed are too small to saturate even an H100. Because B200 devices have even more SMs, and because systems such as NVL72 allow TP factors well beyond eight (up to TP72), the under‑utilization would become even worse. For these reasons, matrices that appear "large" are often still too small to utilize hardware efficiently—making FFN fusion appealing, especially on future GPUs.
>
> If we misunderstood your question or if our explanation didn’t fully address it, we’d be happy to clarify further.
>
> *”The majority of the performance gains come from the existing Puzzle method rather than the proposed Fusion technique.”*
>
> We agree that most performance gains for 253B-Fusion were achieved by Puzzle. Importantly, Puzzle and FFN Fusion are complementary techniques—their benefits stack rather than compete. Puzzle is a collection of efficiency methods (including channel pruning, layer removal, operation replacements like GQA, etc.), and FFN Fusion adds another orthogonal tool to the toolbox. Also, as noted earlier, the speedup from FFN Fusion can increase with stronger GPUs or larger TP configurations.
>
> Looking ahead and after recognizing FFN Fusion’s potential benefits, one could also consider architectural decisions and NAS strategies to further increase the potential benefits:
>
> (a) Longer FFN sequences could be promoted using NAS techniques like Puzzle. The current 253B model was produced using Puzzle’s default settings. However, Puzzle could be adapted to favor attention removals that later enable fusion—making FFN Fusion even more impactful. In fact, FFN Fusion could be introduced as an “alternative block” type within Puzzle, allowing the MIP solver to jointly consider attention removal and FFN fusion during optimization. We have begun preliminary experiments in this direction.
>
> (b) Most current LLMs are built with homogeneous transformer blocks, each consisting of an attention layer and an FFN layer. However, future models could be designed with sequential FFN layers from the start, increasing the number of FFN sequences that could potentially be fused. This could even benefit model performance, as demonstrated in [1].
>
>
> [1] Improving Transformer Models by Reordering their Sublayers, Ofir Press et al., ACL 2020.

---

### Official Review · Reviewer_zYpm · 2025-06-29

**Clarity:** 4
**Significance:** 4
**Originality:** 4
**Rating:** 5
**Confidence:** 4

**Summary:**

This paper proposes FFN Fusion, a novel optimization method that removes attention layers from transformer-based LLMs and fuses sequential FFNs so that they are processed in parallel, leading to significant memory and latency improvements. The method operates by leveraging cosine distance to identify the preceding blocks of which a given block depends on. It then removes the attention blocks of contiguous blocks with low inter-dependency and fuses the remaining sequential FFN layers through a theorem that leverages the use of SwiGLU in modern LLMs. Empirical results show that this method has a relatively minimal performance impact as compared to the original model, and additional post-application finetuning leads to even stronger performance. Ablations show that removing the FFN layers entirely  can severely impact performance and that reversing the FFN layers has little impact, suggesting that the small magnitudal and directional changes of each FFN are not inter-dependent.

**Questions:**

- Is the proposed method effective for models outside of the Llama family?
- what was the time/compute usage for the experiments?
- how does the proposed method par against existing quantization/pruning methods?
- Do we still see similar patterns of inter-block dependency with smaller models?

I would be happy to raise my score to Accept if these questions are effectively and thoroughly addressed.

**Ethical Concerns:**

["NO or VERY MINOR ethics concerns only"]

**Final Justification:**

The authors sufficiently addressed all of my concerns, of which there were many, with strong empirical results and analysis. The proposed method is novel and strongly backed by empirical results. It adds a new compression technique that is both complementary and combinable with existing techniques in the field, making it both relevant and practical.

**Limitations:**

yes

**Quality:**

2

**Strengths And Weaknesses:**

Strengths:

- The initial empirical results strongly support the effectiveness of the method
- The method is applicable to modern LLMs, for which there is a significant need for techniques of this nature
- the work conducts ablations which both demonstrate that the FFNs are still needed and provide explanation for why the method works
- the paper is generally well written with a clear motivation for why the method is useful

Weaknesses:

- it appears that only two models were evaluated, and they both came from the Llama family.  It is therefore unclear whether this method will generalize to models outside of the Llama family that use SwiGLU.
- There is no discussion on how much time/compute was used for the pairwise block dependency calculations
- While the motivation discusses alternate methods like quantization and pruning, there is no discernable comparison to techniques from these fields. It would be nice to see these comparisons. Perhaps the best technique from each field which can equivalently bring down the memory/latency as FFN Fusion does, and how the performance compares.
- While 70B and 405B are the most relevant model sizes for this technique, it would have been interesting to see even smaller sizes and whether or not the insights found for the larger sizes generalize to smaller ones. Because there is no discussion of time/compute needed for this method, I am not sure if this is a reasonable request for the discussion period

---

> ### Author Rebuttal · Authors · 2025-07-30
>
> Thank you for your detailed feedback,
>
> *”it appears that only two models were evaluated, and they both came from the Llama family.”*
>
> Point well taken. During the rebuttal period, we added experiments on Llama 8B 3.1 (see below) from the same family, as well as applying FFN fusion to Mistral Large 2, expanding beyond the Llama family. The process for the Mistral experiment was as follows:
>
> (1) **Applying the Puzzle algorithm** to remove attention layers. We removed 27 layers that resulted in 2 sequences of 14 and 13 consecutive FFN layers.
>
> (2) **Applying FFN Fusion** to fuse all but the final FFN layer in each sequence.
>
> We did not have time during the rebuttal to run any recovery training, but for the revision we will also include results with KD.
> Looking at the final results, the fused model retained 99.4% of the pre-fusion MMLU score and 94.4% of its MT-Bench score.
> We also intend to include models from additional diverse families for the revision.
>
>
>
> *”it would have been interesting to see even smaller sizes and whether or not the insights found for the larger sizes generalize to smaller ones”*
>
> To determine if FFN Fusion is effective for smaller models, we tested it on the Llama 8B 3.1 Instruct model. The process involved:
>
> (1) **Applying the Puzzle algorithm** to remove attention layers. We removed 9 layers, resulting in 8 consecutive FFN layers—a slightly smaller proportion compared to the 70B case.
>
> (2) **Applying FFN Fusion** to fuse all but the final FFN layer in this sequence.
>
> (3) **Running a short KD recovery phase** over 20B tokens.
>
> Compared to the original (unmodified) Llama 8B model, the final fused model retained 98.99% / 96.5% of its performance on MMLU / MT-Bench, respectively. This confirms that FFN Fusion is effective at smaller scales as well.
>
> *”Do we still see similar patterns of inter-block dependency with smaller models?”*
>
> That’s a good question. Beyond the fusion experiment for Llama 8B 3.1 Instruct, following your question, we also analyzed the model's inter-block dependency. We observed that, like the 253B and 49B models, the fused layers showed low dependency, while other layers were more dependent on their neighbors. We aim to add the plot to the revised version alongside the corresponding fusion results.
>
> *”There is no discussion on how much time/compute was used for the pairwise block dependency calculations”*
>
> *”what was the time/compute usage for the experiments?”*
>
> Calculating the pairwise block dependencies took 0.24 H100 GPU hours per matrix row for Llama 405B.
> It’s important to clarify that this step was part of an exploratory experiment and is not required for applying FFN Fusion itself. The fusion step is extremely lightweight—it involves direct merging of FFN matrices and does not require any heavy computation. This is why we did not report compute usage for it. The block dependency analysis is only relevant to the Block Parallelization method discussed in Appendix B.
> Other compute-intensive steps, such as KD training or CPT, are detailed in Section 4 under “Additional Training”, where we provide the full token budgets.
> Are there any other time/compute steps you think we should also include in the revision?
>
>
> *”While the motivation discusses alternate methods like quantization and pruning, there is no discernable comparison to techniques from these fields. It would be nice to see these comparisons.”*
>
> Our paper presents FFN Fusion as a technique complementary to existing inference efficiency methods like pruning and quantization. Pruning improves matrix multiplication speed by leveraging sparsity, while FFN Fusion reduces synchronization overhead by decreasing the number of computational blocks.
> Notably, the models in this paper already combine both approaches: FFN channel pruning (applied during Puzzle) and FFN Fusion, showcasing the benefits of their combined application. Similarly, all results in Section 4 are shown with FP8 quantization. Importantly, our experiments (detailed in Section 5.2) already compare FFN Fusion to pruning entire FFN layers, which we feel is the best baseline for our method, and have demonstrated the superior performance of our approach.
>
> However, to offer another comparison between FFN Fusion and pruning, we ran an experiment on the 49B model from Section 5. We applied structured sparsity (Wanda [1]) to the FFN weights, achieving 40% sparsity. The resulting model had a latency profile similar to “step 3” in Table 3. The sparse model scored 78.26 MMLU, while our fused model achieved 80.36 MMLU. As a next step—also planned for inclusion in the revision—we aim to explore combining both approaches.
>
> [1] A Simple and Effective Pruning Approach for Large Language Models, Sun et al.
>
>
> We hope we have thoroughly addressed your questions.

---

> > ### Comment · Reviewer_zYpm · 2025-08-01
> >
> > I thank the authors for their comprehensive response. I would suggest that, in addition to the GPU hours per matrix row, they include the total GPU hours per model (so the reader doesn't have to perform the napkin math). Furthermore, if you are also combining with WANDA, you may want to combine with some SOTA quantization technique as an interesting experiment for future revision [1]. Other than that, all of my concerns have been satisfied. I will raise my score to Accept.
> >
> > [1] Tseng, Albert, et al. ‘QTIP: Quantization with Trellises and Incoherence Processing’. Advances in Neural Information Processing Systems, edited by A. Globerson et al., vol. 37, Curran Associates, Inc., 2024, pp. 59597–59620, https://proceedings.neurips.cc/paper_files/paper/2024/file/6de2e84b8da47bb2eb5e2ac96c63d2b0-Paper-Conference.pdf.

---

> > > ### Author Response · Authors · 2025-08-05
> > > **Comment**
> > >
> > > We appreciate your suggestion; for clarity, the total GPU hours were 5.4 H100 hours for Llama 70B (0.067~ hours per row), and 30.24 H100 hours for Llama 405B (0.24 hours per row). We will include these numbers in the revision for the readers’ convenience.
> > >
> > > Thank you as well for the recommendation regarding quantization techniques. We will consider QTIP, along with several other methods we've discussed internally, as part of our future revision.

---

### Official Review · Reviewer_Pynj · 2025-07-01

**Clarity:** 3
**Significance:** 2
**Originality:** 3
**Rating:** 5
**Confidence:** 3

**Summary:**

This paper identifies inference inefficiencies of recent Mixture of Expert architectures for small and medium batch sizes due to the multiple small FFN (sub-)modules (i.e. experts).
Instead, the paper improves hardware utilization during inference by combining FFN layers into single wider layers after training, thereby reducing the serial computation (due to the number of layers) and increasing the parallel computation.
Using the proposed method FFN Fusion, the paper introduces a 253B parameter dense model derived from Llama-3.1 405B, which shows a 1.71x inference speed up at equal or higher quality compared to the base model.

**Questions:**

- L. 168: For TP can‘t we overlap computation and communication and in this way hide the communication time?
- Section E: The authors only talk about „single token inference“, so I assume the authors mean the inefficiency during generation (?), but it is not mentioned. How is it for prefill?
- Section E: What are intermediate / large batch sizes in this case? The motivation would be much stronger with (a) concrete example(s).
- Section E: What is meant with suboptimal and optimal region of efficiency? The explanation is rather vague.

- Theorem 3.1 shows that the parallel execution is identical to the parallel execution of FFN layers. However, does it also hold in the presence of skip connections and norm layers around the sequential FFN layers?

- Figure 2: Why is there a significant difference in the columns 110 - 120 compared to the earlier layers?
- How do you compute the cosine distance over matrices n x d ? Do you average over the sequence dimension?

Section 5.3: Do you have an explanation why the last layer in a sequence of blocks is especially important?

**Ethical Concerns:**

["NO or VERY MINOR ethics concerns only"]

**Final Justification:**

The authors provided new insightful experiments, which address my concerns on generality (e.g. smaller models and/or models from other family) and they also clarified their motivation (provided in Appendix E).

**Limitations:**

yes

**Paper Formatting Concerns:**

No conerns.

**Quality:**

3

**Strengths And Weaknesses:**

Strengths:
- FFN Fusion yields higher inference throughput at equal or better performance than the base model.
- Thorough empirical investigation (even though only at very large model sizes (70B and 253B)).

Weaknesses:
- The motivation of inefficient mixture of experts for small batch sizes would be stronger if the authors could give a concrete example including latency and throughput calculation or measurements, which demonstrate the weaknesses of Mixture of Experts (See also the questions on Section E).
- The experiments are performed only on two models from the same family, which likely used the same training recipe and training dataset. While the authors provide thorough empirical analyses on these two models, it still leaves the question of how general this method is. Also, FFN fusion has only been applied to very large models.
- Experiments at smaller scales (e.g. 8B - 32B), where there are more open source models available, could further prove the generality of FFN Fusion.
- Figure 5: To really explore the efficiency frontier more models should be added to this plot (potentially including MoE models, to really see the difference in Latency, see also first point)

---

> ### Author Rebuttal · Authors · 2025-07-30
>
> Thank you for your feedback and helpful questions regarding Appendix E.
>
> *”L. 168: For TP can‘t we overlap computation and communication and in this way hide the communication time?”*
>
> TP does not enable communication hiding. The all-reduce (summation) operation at the end of the TP cycle requires all GPUs to complete their computation before communication can begin.
> In theory, overlapping computation and communication is possible by partitioning data (activations) in some way. However, this is only practical through micro-batching, which introduces its own challenges. A fine-grained approach—such as streaming portions of the output as they become ready—shows little to no benefit when the data volume is small.
>
> *”Section E: The authors only talk about „single token inference“, so I assume the authors mean the inefficiency during generation (?), but it is not mentioned. How is it for prefill?”*
>
> While the MoE analysis focuses on generation, the key factor for inference performance here is not batch size but the number of tokens in the forward pass. In that sense, prefill behaves similarly to generation when the number of tokens is the same. For example, a prefill with 256 tokens has similar behavior to generation with a batch size of 256. So the analysis in Appendix E applies in both cases.
>
> *”Section E: What are intermediate / large batch sizes in this case? The motivation would be much stronger with (a) concrete example(s).”*
>
> *”Section E: What is meant with suboptimal and optimal region of efficiency? The explanation is rather vague.”*
>
> Your questions helped us clarify Appendix E, and we will include the following explanation in the revision:
>
> **Background: Linear layers.**
>  The performance of linear layers varies significantly with batch size. For small batches, performance is limited by the time needed to read weights from DRAM (I/O bottleneck). For large batches, performance becomes compute-bound, limited by the hardware's maximal FLOPs/sec. Notably, the latency of a linear layer remains the same for batch sizes 1 and 64, as both are dominated by weight loading time. This makes small batches extremely inefficient. While other factors exist, we consider a linear layer to operate efficiently once the number of tokens in the batch approaches 64, and inefficient otherwise.
>
> **Back to MoEs:**
> As mentioned in Appendix E, MoEs require all experts to be active for high throughput. However, the total batch is split across experts, significantly reducing the effective number of tokens each expert sees. For example, with a sparsity factor of 1/32 (as in DeepSeek-V3\R1), a batch of 2048 tokens (64x32) is needed to ensure each expert processes around 64 tokens. The case is more extreme in Llama 4 Maverick, where a 1/128 sparsity factor requires a batch of 8192 tokens (64x128) to reach this threshold. These batch sizes are often impractical—especially in generation mode—leaving MoEs to operate in a low-efficiency regime, where linear layers are under-utilized.
> We loosely define the *efficient* or *optimal* zone as having ≥64 tokens per expert; below that, operation enters a *suboptimal* or *inefficient* region due to underutilized linear layers.
>
> *”The experiments are performed only on two models from the same family”*
>
> Point well taken. During the rebuttal period, we added experiments on Llama 8B 3.1 (see below) from the same family, as well as applying FFN fusion to Mistral Large 2, expanding beyond the Llama family. The process for the Mistral experiment was as follows:
>
> (1) **Applying the Puzzle algorithm** to remove attention layers. We removed 27 layers that resulted in 2 sequences of 14 and 13 consecutive FFN layers.
>
> (2) **Applying FFN Fusion** to fuse all but the final FFN layer in each sequence.
>
> We did not have time during the rebuttal to run any recovery training, but for the revision we will also include results with KD.
> Looking at the final results, the fused model retained 99.4% of the pre-fusion MMLU score and 94.4% of its MT-Bench score.
> We also intend to include models from additional families for the revision.
>
>
>
> *”Experiments at smaller scales (e.g. 8B - 32B), … could further prove the generality of FFN Fusion.”*
>
>
> To determine if FFN Fusion is effective for smaller models, we tested it on the Llama 8B 3.1 Instruct model. The process involved:
>
> (1) **Applying the Puzzle algorithm** to remove attention layers. We removed 9 layers, resulting in 8 consecutive FFN layers—a slightly smaller proportion compared to the 70B case.
>
> (2) **Applying FFN Fusion** to fuse all but the final FFN layer in this sequence.
>
> (3) **Running a short KD recovery phase** over 20B tokens.
>
> Compared to the original (unmodified) Llama 8B model, the final fused model retained 98.99% / 96.5% of its performance on MMLU / MT-Bench, respectively. This confirms that FFN Fusion is effective at smaller scales as well.
>
> *”Figure 5: To really explore the efficiency frontier more models should be added to this plot (potentially including MoE models, to really see the difference in Latency, see also first point)”*
>
>  We agree, and plan to add models such as Mistral and Deepseek in the revision.
>
>
> *”Theorem 3.1 shows that the parallel execution is identical to the parallel execution of FFN layers. However, does it also hold in the presence of skip connections and norm layers around the sequential FFN layers?”*
>
> Theorem 3.1 shows that Equation (5), which describes a parallel execution of FFN layers with shared normalization parameters, is mathematically equivalent to a single denser FFN layer using the same normalization parameters. Our main empirical observation is that in trained LLMs, sequences of FFN layers without intermediate attention—though each with their own skip connections and normalization parameters—can still be replaced with the parallel form in Equation (5). As you noted, while this substitution from a sequential to a parallel architecture isn't mathematically equivalent, our results prove it to maintain accuracy while improving efficiency.
>
> *”Figure 2: Why is there a significant difference in the columns 110 - 120 compared to the earlier layers?”*
>
>
> We observed a distinct change in dependency patterns between the fused FFN layers (layers 66–114) and the subsequent layers (114–120). Unlike the earlier sequence, the later layers retain their attention modules (they were not removed by Puzzle) and show stronger dependencies along the main diagonal.
>
> While we don’t have a definitive explanation for this phenomenon ([1] attempts to characterize different functional stages of the network), this increased dependency seems to make the later layers harder to modify. Although we lack direct evidence, we hypothesize that these final layers play a key role in shaping the token representations for the LM head and final token prediction.
>
> *”How do you compute the cosine distance over matrices n x d ? Do you average over the sequence dimension?”*
>
> Yes, we compute the cosine distance across the hidden dimension for each token, and then average the results over the sequence dimension.
>
>
> *”Section 5.3: Do you have an explanation why the last layer in a sequence of blocks is especially important?”*
>
> That’s an interesting question. While we don’t have a definitive answer, we hypothesize that the final FFN in a sequence may serve a special role. In our case, Puzzle chose not to remove the attention layer following it—possibly indicating that this attention depends on the output of the preceding (non-fused) FFN.
>
> [1] The Remarkable Robustness of LLMs: Stages of Inference?, Lad et al.

---

> > ### Comment · Reviewer_Pynj · 2025-08-04
> >
> > Thanks for this extensive rebuttal and the clarifications to my questions.
> > The authors provided new insightful experiments, which address my concerns on generality (e.g. smaller models and/or models from other family).
> > I will raise my score to accept.

---

### Official Review · Reviewer_RXRj · 2025-07-07

**Clarity:** 3
**Significance:** 3
**Originality:** 4
**Rating:** 4
**Confidence:** 5

**Summary:**

This paper presents FFN fusion, a post-training architecture optimization technique that removes attention layers from certain Transformer blocks and fuses the remaining feedforward layers into one. The method is novel and sheds light on the effectiveness of individual attention/feedforward layers in a pretrained LLaMA model. The downstream performance along with inference speedup appears promising.

**Questions:**

To what extent do you think the current method can generalize? One concern I have is the normalization strategy. Specifically, the usage of pre-norm might affect many design choices in this paper. For example, the h(x) function used in the pairwise block dependency seems to depend on the pre-norm structure. In cases with other normalization structures like post-norm, can we still remove attention layers and merge FFN layers effectively?

**Ethical Concerns:**

["NO or VERY MINOR ethics concerns only"]

**Limitations:**

Strong performance with unclear generalization across different architectures and settings.

**Quality:**

3

**Strengths And Weaknesses:**

### Strength
1. Well-motivated and well-designed approach that provides valuable insights into layer-wise contributions in pre-trained models.
2. Strong performance demonstrating improvements in both quality and efficiency metrics.

### Weaknesses:
My main concern lies in the generalization of this method:

1. It's unclear how the method can generalize to MoE models, which are standard architectures for very large models.
2. The coupling between the method and LLaMA-specific design choices remains unclear (see question below).
3. The impact of removing attention layers on long-context performance is not addressed, though attention layers are presumably more crucial in long-context settings.

---

> ### Author Rebuttal · Authors · 2025-07-31
>
> Thank you for your constructive feedback,
>
> *”The impact of removing attention layers on long-context performance is not addressed, though attention layers are presumably more crucial in long-context settings.”*
>
> This is an important point, and it is partially addressed in Figure 4, showing that the RULER [1] long-context benchmark results at 128K shows that our 253B CPT model, with half the attention layers of the original Llama-405B, achieves an average score across all RULER tests that is 112.85% of Llama-405B’s. In addition, even before any training or knowledge distillation (KD), the FFN-fused model retains 98.7% of Llama-405B’s performance at 64K context and 79.4% at 128K. We will highlight these results in the revised manuscript, to clarify that long-context performance is preserved.
>
> *”One concern I have is the normalization strategy. Specifically, the usage of pre-norm might affect many design choices in this paper. For example, the h(x) function used in the pairwise block dependency seems to depend on the pre-norm structure. In cases with other normalization structures like post-norm, can we still remove attention layers and merge FFN layers effectively?”*
>
>
> Most current LLMs adopt a pre-norm architecture, including models like Llama, Mistral, DeepSeek, Qwen, and Kimi among many others. Our work focuses on this prevalent setting. That said, exploring the applicability of our approach to post-norm architectures is an interesting direction for future work.
>
> While we were unable to conduct a full empirical study during the rebuttal period, we can offer a structural analysis that highlights potential differences. The main distinction is that post-norm applies normalization after the block output $f(X)$, in contrast to the accumulating behavior seen in pre-norm models. In pre-norm architectures, we observe that token representation norms tend to increase significantly across layers, which results in a decreasing cosine distance between $X$ and $f(X)$—except in the final layers. These lower cosine distances appear to correlate with successful attention pruning and FFN fusion.
> In post-norm models, we expect a different pattern of cosine distances due to the normalization applied at the output. This could lead to different behavior in terms of attention removal and fusion. Nonetheless, we believe that the ability to reduce model capacity—either by pruning attention or fusing FFNs—stems primarily from the over-parameterization of LLMs, rather than from specific architectural choices. For instance, [3] show that early layers in post-norm models are typically less effective, which indicates they might also tolerate attention pruning and FFN Fusion.
>
>
> *”It's unclear how the method can generalize to MoE models, which are standard architectures for very large models.”*
>
> This is indeed a very interesting future direction! We have already developed such possible generalizations of the FFN fusion method for MoE architectures. The simplest of these, involving two MoE layers with T total experts, s shared experts, and r routed experts from which we select k chosen experts per layer, proceeds as follows: first, attention is removed between the MoE layers (where applicable); then, the routers are concatenated, producing an output of dimension 2r. A top-K selection then chooses 2k active experts from the list of 2r experts, and the shared experts are fused into a 2x larger shared expert.
> Our preliminary evidence suggests that some pairs of Llama 4 Scout MoE layers can be fused in such a manner without loss of accuracy in MMLU and MT-Bench. We will explore the effectiveness and refinement of these ideas in future work, and will add this point to the concluding remarks section.
>
>
> [1] RULER: What's the Real Context Size of Your Long-Context Language Models? Hsieh et al.
>
> [2] Puzzle: Distillation-Based NAS for Inference-Optimized LLMs, Bercovich et al.
>
> [3] The Curse of Depth in Large Language Models, Sun et al.

---

### Decision · Program_Chairs · 2025-09-17

**Decision:**

Accept (spotlight)

**Comment:**

This paper proposes FFN fusion, an approach that parallelizes sequential layers in a Transformer by removing specific attention layers and running subsequent FFN layers in parallel. The approach reduces inference latency while preserving model behavior, and the authors evaluate it on Llama-3.1-405B.

The reviewers found the problem space important and found the method to be well-motivated and well-designed, with strong empirical results and baselines.

Given the importance of the problem and novelty in the approach, I recommend this paper for accept with oral or spotlight.